# Warning of a forthcoming collapse of the Atlantic meridional overturning circulation

Peter Ditlevsen [1,3] ✉ & Susanne Ditlevsen [2,3] ✉

The Atlantic meridional overturning circulation (AMOC) is a major tipping element in the climate system and a future collapse would have severe impacts on the climate in the North Atlantic region. In recent years weakening in circulation has been reported, but assessments by the Intergovernmental Panel on Climate Change (IPCC), based on the Climate Model Intercomparison Project (CMIP) model simulations suggest that a full collapse is unlikely within the 21st century. Tipping to an undesired state in the climate is, however, a growing concern with increasing greenhouse gas concentrations. Predictions based on observations rely on detecting early-warning signals, primarily an increase in variance (loss of resilience) and increased autocorrelation (critical slowing down), which have recently been reported for the AMOC. Here we provide statistical significance and data-driven estimators for the time of tipping. We estimate a collapse of the AMOC to occur around mid-century under the current scenario of future emissions.

A forthcoming collapse of the Atlantic meridional overturning circulation (AMOC) is a major concern as it is one of the most important tipping elements in Earth's climate system[1–3]. In recent years, model studies and paleoclimatic reconstructions indicate that the strongest abrupt climate fluctuations, the Dansgaard-Oeschger events[4], are connected to the bimodal nature of the AMOC[5,6]. Numerous climate model studies show a hysteresis behavior, where changing a control parameter, typically the freshwater input into the Northern Atlantic, makes the AMOC bifurcate through a set of co-dimension one saddlenode bifurcations[7–9]. State-of-the-art Earth-system models can reproduce such a scenario, but the inter-model spread is large and the critical threshold is poorly constrained[10,11]. Based on the CMIP5 generation of models, the AR6 IPCC report quotes a collapse in the 21st century to be very unlikely (medium confidence)[12]. Among CMIP6 models, there is a larger spread in the AMOC response to warming scenarios, thus an increased uncertainty in the assessment of a future collapse[13]. There are, however, model biases toward overestimated stability of the AMOC, both from tuning to the historic climate record[14], poor representation of the deep water formation[15], salinity and glacial runoff[16].

When complex systems, such as the overturning circulation, undergo critical transitions by changing a control parameter $\lambda$ through

a critical value $\lambda_c$, a structural change in the dynamics happens. The previously statistically stable state ceases to exist and the system moves to a different statistically stable state. The system undergoes a bifurcation, which for $\lambda$ sufficiently close to $\lambda_c$ can happen in a limited number of ways rather independent from the details in the governing dynamics[17]. Besides a decline of the AMOC before the critical transition, there are early-warning signals (EWSs), statistical quantities, which also change before the tipping happens. These are critical slowing down (increased autocorrelation) and, from the Fluctuation-Dissipation Theorem, increased variance in the signal[18–20]. The latter is also termed "loss of resilience", especially in the context of ecological collapse[21]. The two EWSs are statistical equilibrium concepts. Thus, using them as actual predictors of a forthcoming transition relies on the assumption of quasi-stationary dynamics.

The AMOC has only been monitored continuously since 2004 through combined measurements from moored instruments, induced electrical currents in submarine cables and satellite surface measurements[22]. Over the period 2004–2012, a decline in the AMOC has been observed, but longer records are necessary to assess the significance. For that, careful fingerprinting techniques have been applied to longer records of sea surface temperature (SST), which, backed by a survey of a large ensemble of climate model simulations,

[1]Niels Bohr Institute, University of Copenhagen, Copenhagen, Denmark. [2]Institute of Mathematical Sciences, University of Copenhagen, Copenhagen, Denmark. [3]These authors contributed equally: Peter Ditlevsen, Susanne Ditlevsen. ✉e-mail: pditlev@nbi.ku.dk; susanne@math.ku.dk

have found the SST in the Subpolar gyre (SG) region of the North Atlantic (area marked with a black contour in Fig. 1a) to contain an optimal fingerprint of the strength of the AMOC[23–25].

Figure 1b shows the SG and the GM SSTs obtained from the Hadley Centre Sea Ice and Sea Surface Temperature data set (HadISST)[26]. Figure 1c shows the SG anomaly, and Fig. 1d shows the GM anomaly with a clear global warming trend in the last half of the record. The AMOC fingerprint for the period 1870–2020 is shown in Fig. 1e. This is the basis for the analysis. It has been reported[11,27] that this and similar AMOC indices show significant trends in the mean, the variance and the autocorrelation, indicating early warning of a shutdown of the AMOC. However, a trend in the EWSs within a limited period of observation could be a random fluctuation within steady-state statistics. Thus, for a robust assessment of the shutdown, it is necessary to establish a statistical confidence level for the change above the natural fluctuations. This is not easily done given only one, the observed realization of the approach to the transition. Here we establish such a measure of the confidence for the variance and autocorrelation and demonstrate that variance is the more reliable of the two. A further contribution is an estimator of not only whether a transition is approaching but also the time when the critical transition is expected to occur. The strategy is to infer the evolution of the AMOC solely on observed changes in mean, variance and autocorrelation. The typical choice of control parameter is the flux of freshwater into the North Atlantic. River runoff, Greenland ice melt and export from the Arctic Ocean are not well constrained[28]; thus, we do not assume the control parameter known. Boers[27] assumes the global mean temperature $T$ to represent the control parameter. Although $T$ has increased since ~1920 (Fig. 1d), the increase is not quite linear with time. All we assume here is that the AMOC is in an equilibrium state prior to a change toward the transition. The simplest uninformed assumption is that the change is sufficiently slow and that the control parameter approaches the (unknown) critical value linearly with time. This assumption is confirmed by a close fit of the estimated model to the observed AMOC fingerprint. Although we make no explicit assumptions, the primary driver of climate change, the logarithm of the atmospheric $CO_2$ concentration, does, in fact, increase close to linearly with time in the industrial period[29]. Our results are robust without making specific assumptions regarding the driver of the AMOC.

In this work, we show that a transition of the AMOC is most likely to occur around 2037–2109 (95% confidence interval).

## Results

### Modeling and detecting the critical transition

Denote the observed AMOC fingerprint by $x(t)$ (Fig. 1e). We model it by a stochastic process $X_t$, which, depending on a control parameter $\lambda < 0$, is at risk of undergoing a critical transition through a saddle-node bifurcation for $\lambda = \lambda_c = 0$. The system is initially in a statistically stable state, i.e., it follows some stationary distribution with constant $\lambda = \lambda_0$. We are uninformed about the dynamics governing the evolution of $X_t$ but can assume effective dynamics, which, with $\lambda$ sufficiently close to the critical value $\lambda_c = 0$, can be described by the stochastic differential equation (SDE):

$$dX_t = -(A(X_t - m)^2 + \lambda)dt + \sigma dB_t, \qquad (1)$$

where $m = \mu - \sqrt{|\lambda|/A}$ and $\mu$ is the stable fixed point of the drift, $A$ is a time scale parameter, $B_t$ is a Brownian motion and $\sigma^2$ scales the variance. Disregarding the noise, this is the normal form of the co-dimension one saddle-node bifurcation[17] (see "Methods"). The square-root dependence of the stable state: $\mu - m \sim \sqrt{\lambda_c - \lambda}$ is the main signature of a saddle-node bifurcation. It is observed for the AMOC shutdown in ocean-only models as well as in coupled models, see Fig. 2, in strong support of Eq. (1) for the AMOC.

At time $t_0$, $\lambda(t)$ begins to change linearly toward $\lambda_c = \lambda(t_c) = 0$:

$$\lambda(t) = \lambda_0(1 - \Theta[t - t_0](t - t_0)/\tau_r), \qquad (2)$$

where $\Theta[t]$ is the Heaviside function and $\tau_r = t_c - t_0 > 0$ is the ramping time up to time $t_c$, where the transition eventually will occur. Time $t_c$ is denoted the tipping time; however, an actual tipping can happen earlier than $t_c$ due to a noise-induced tipping. As the transition is approached, the risk of noise-induced tipping (n-tipping) prior to $t_c$ is increasing and, at some point, making the EWSs irrelevant for predicting the tipping. The probability for n-tipping can, in the small noise limit, be calculated in closed form, $P(t,\lambda) = 1 - \exp(-t/\tau_n(\lambda))$, with mean waiting time $\tau_n(\lambda) = (\pi/\sqrt{|\lambda|}) \exp(8|\lambda|^{3/2}/3\sigma^2)$ (see "Methods").

The mean and variance are calculated from the observations as the control parameter $\lambda(t)$ is possibly changing. These EWSs are inherently equilibrium concepts and statistical; thus, a time window, $T_w$, of a certain size is required for a reliable estimate. As the transition is approached, the differences between the EWSs and the pre-ramping values of the variance and autocorrelation (baseline) increase; thus, a shorter window $T_w$ is required for detecting a difference. Conversely, close to the transition critical slowing down decreases the number of independent points within a window, thus calling for a larger window for reliable detection. Within a short enough window, $[t - T_w/2, t + T_w/2]$, we may assume $\lambda(t)$ to be constant and the noise small enough so that the process (1) for given $\lambda$ is well approximated by a linear SDE, the Ornstein–Uhlenbeck process[30]. A Taylor expansion around the fixed point $\mu(\lambda)$ yields the approximation

$$dX_t \approx -\alpha(\lambda)(X_t - \mu(\lambda))dt + \sigma dB_t \qquad (3)$$

where $\mu(\lambda) = m + \sqrt{|\lambda|/A}$ and $\alpha(\lambda) = 2\sqrt{A|\lambda|}$ is the inverse correlation time. For fixed $\lambda$, the process is stationary, with mean $\mu$, variance $\gamma^2 = \sigma^2/2\alpha$ and one-lag autocorrelation $\rho = \exp(-\alpha\Delta t)$ with step size $\Delta t = 1$ month. As $\lambda(t)$ increases, $\alpha$ decreases, and thus variance and autocorrelation increase. From $\mu$, $\gamma^2$ and $\rho$ the parameters of Eq. (1) are determined: $\alpha = -\log\rho/\Delta t$, $\sigma^2 = 2\alpha\gamma^2$, $A = \alpha/2(\mu - m)$ and $\lambda = (\sigma^2/4\gamma^2)^2/A$. Closed form estimators for $\mu$, $\gamma^2$ and $\rho$ are obtained from the observed time series within a running window by maximum likelihood estimation (MLE) (Supplementary text S1, see also ref. 31).

The uncertainty is expressed through the variances of the estimators $\hat{\gamma}^2$ and $\hat{\rho}$ obtained from the observations within a time window $T_w$. The hats indicate that they are estimators and thus stochastic variables with variances around the true values. Detection of an EWS at some chosen confidence level $q$ (such as 95 or 99%) requires one of the estimates $\hat{\gamma}^2$ or $\hat{\rho}$ for a given window to be statistically different from the baseline values $\hat{\gamma}_0^2$ or $\hat{\rho}_0$, which depend on the window size as well as how different the EWSs are from their baseline values.

### Time scales in early-warning signals

The detection of a forthcoming transition using statistical measures involves several time scales. The primary internal time scale is the autocorrelation time, $t_{ac}$, in the steady state. The ramping time $\tau_r$ over which the control parameter changes from the steady state value to the critical value sets an external time scale. For given $\alpha(\lambda)$ and $q$-percentile, the required time window $T_w(q, \alpha)$ to detect a change from baseline in EWSs at the given confidence level $q$ is given in the closed form in the next section (Eq. (7) for variance and Eq. (8) for autocorrelation). The approach to the collapse and the involved time scales are schematically summarized in Fig. 3, while they are calculated in Fig. 4a, where the required window size $T_w$ at the 95% confidence level is plotted as a function of $\lambda$ for the variance (red curve) and autocorrelation (yellow curve). These are plotted together with the mean

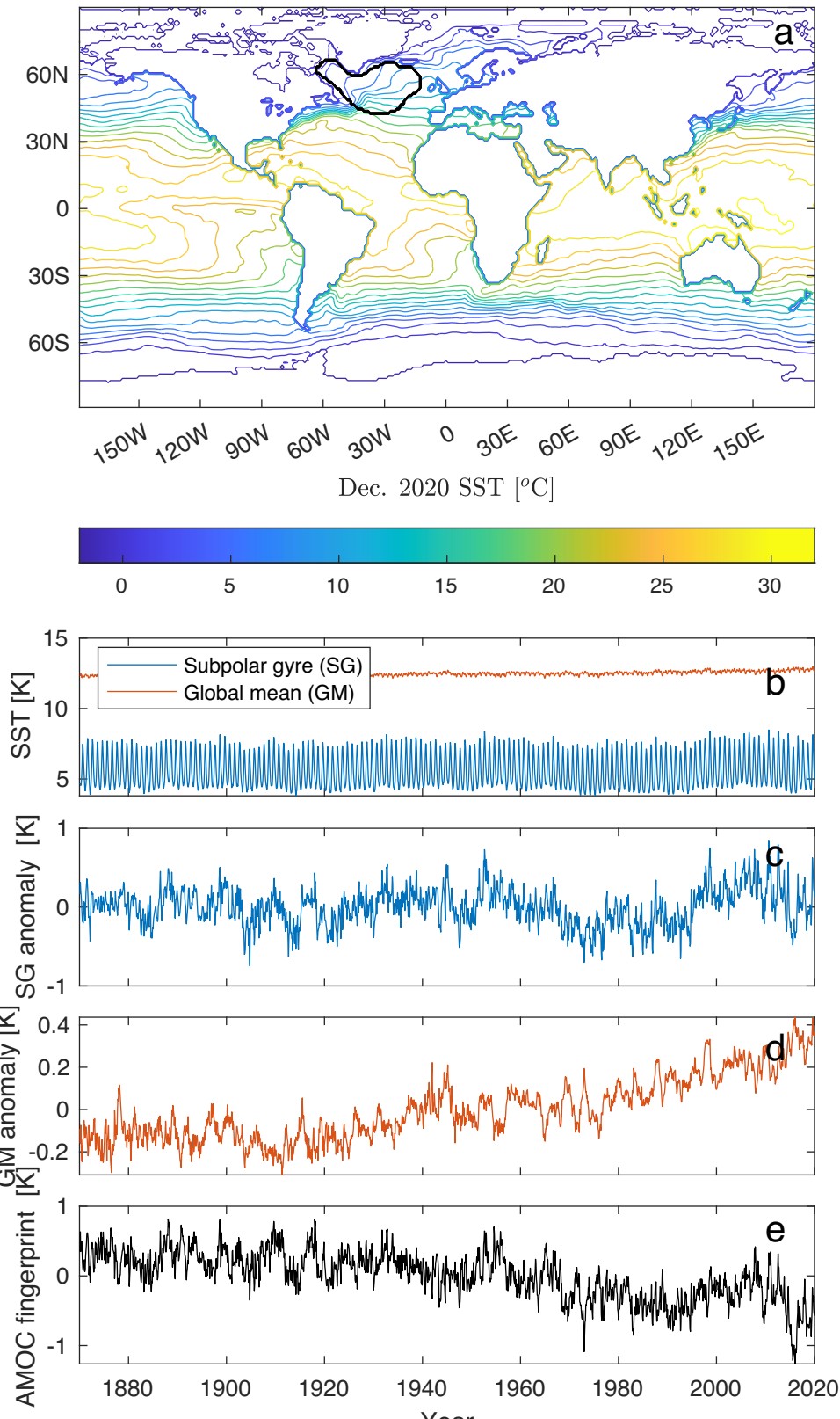

**Fig. 1 | The Atlantic meridional overturning circulation (AMOC) fingerprint, sea surface temperature (SST) and global mean (GM). a** Subpolar gyre (SG) region (black contour) on top of the Hadley Centre Sea Ice and Sea Surface Temperature data set (HadISST) SST reconstruction for December 2020. The SG region SST has been identified as an AMOC fingerprint[23]. **b** Full monthly record of the SG SST together with the global mean (GM) SST. **c, d** SG and GM anomalies, which are the records subtracted the monthly mean over the full record. **e** AMOC fingerprint proxy, which is here defined as the SG anomaly minus twice the GM anomaly, compensating for the polar amplified global warming.

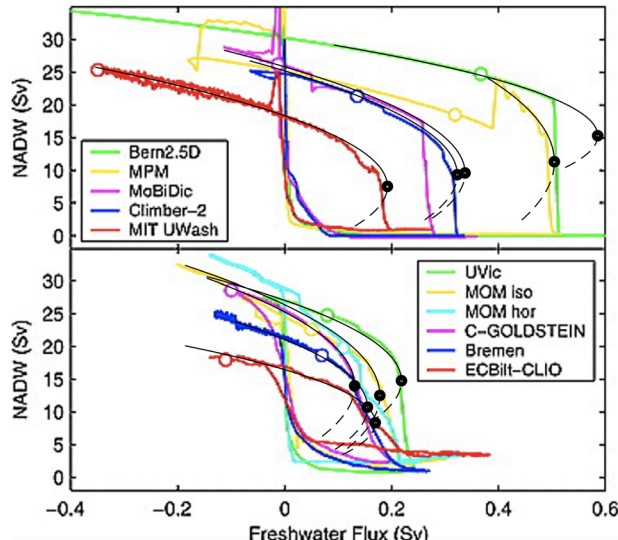

**Fig. 2 | Steady state curves from climate model simulations of the North Atlantic Deep Water (NADW), with a very slowly changing control parameter (freshwater forcing).** Top panel shows ocean-only models, while bottom panel shows atmosphere-ocean models. The curves are, even away from the transition, surprisingly well fitted by Eq. (1) (black thin curves). The bifurcation points are indicated with black circles. Note that for some models, the transition happens before the critical point, as should be expected from noise-induced transitions. The colored circles show the present-day conditions for the different models. Adapted from Rahmstorf et al.[34].

waiting time for n-tipping, $t_{noise}$, (blue curve). With $T_w = 50$ years, increased variance can only be detected after the time when $\lambda(t) \approx -1.2$ (crossing of red and red-dashed curves). At that time, a window of approximately 75 years is required to detect an increase in autocorrelation, making variance the better EWS of the two. When $\lambda \approx -0.4$, the mean waiting time for n-tipping is smaller than the data window size. Thus, the increased variance can be used as a reliable EWS in the range $-1.2 < \lambda(t) < -0.4$, indicated by the green band. How timely an early warning this is depends on the speed at which $\lambda(t)$ is changing from $\lambda_0$ to $\lambda_c$, i.e., the ramping time $\tau_r$. A set of 1000 realizations has been simulated with $\lambda_0 = -2.82$ and $\tau_r = 140$ years, indicated by the time labels on top of Fig. 4a. Ten of these realizations are shown in Fig. 4b on top of the stable and unstable branches of fixed points of model (1) (the bifurcation diagram). Figure 4c (d) shows the variance (autocorrelation) calculated from the realizations within a running 50-year window (shown in Fig. 4c). The solid black line is the baseline value for $\lambda = \lambda_0$, while the solid blue line is the increasing value for $\lambda = \lambda(t)$. The calculated 95% confidence level for the measurement of the EWS within the running window is shown by the dashed black and blue lines, respectively. The corresponding light blue curves are obtained numerically from the 1000 realizations. The green band in Fig. 4c corresponds to the green band in Fig. 4a and shows where early warning is possible in this case.

**Statistics of early-warning signals**
The variances of the estimators are approximately (see Supplementary text S1).

$$\text{Var}(\hat{\gamma}^2) \approx \frac{2(\gamma^2)^2}{\alpha T_w} = \frac{\sigma^4}{2\alpha^3 T_w}; \ \text{Var}(\hat{\rho}) \approx \frac{2\alpha \Delta t^2}{T_w}, \qquad (4)$$

where $T_w = n\Delta t$ is the observation window.

The question is then how large $T_w$ needs to be to detect a statistically significant increase compared to the estimated baseline values $\hat{\gamma}_0^2$ and $\hat{\rho}_0$. For a given estimate $\hat{\gamma}^2$, the estimated difference from the

baseline variance is

$$\Delta_{\gamma^2} = \hat{\gamma}^2 - \hat{\gamma}_0^2 = \hat{\gamma}_0^2(\hat{\alpha}_0/\hat{\alpha} - 1), \qquad (5)$$

and the estimated difference from the baseline autocorrelation is

$$\Delta_\rho = \hat{\rho} - \hat{\rho}_0 = \hat{\rho}_0(e^{(\hat{\alpha}_0 - \hat{\alpha})\Delta t} - 1) \approx \hat{\rho}_0(\hat{\alpha}_0 - \hat{\alpha})\Delta t. \qquad (6)$$

Since the two EWSs, $\hat{\gamma}^2$ and $\hat{\rho}$, are treated on an equal footing, in the following, we let $\hat{\psi}$ denote either of the estimators (given explicitly in Supplementary text S1, Eqs. (S5) or (S6)). The standard error is $s(\hat{\psi}) = \text{Var}(\hat{\psi})^{1/2}$ (Eq. (4)) and $\hat{\Delta}$ denotes either of the two estimated differences (5) or (6). The null hypothesis is that $\lambda = \lambda_0$, or equivalently $\alpha = \alpha_0$. The null distribution of $\hat{\psi}$ is assumed to be Gaussian (confirmed by simulations). A quantile $q$ from the standard Gaussian distribution expresses the acceptable uncertainty in measuring the statistical quantity $\psi$. We thus get that $\hat{\Delta} < qs(\hat{\psi})$ at the $q$-confidence level (95%, 99% or similar) under the null hypothesis. To detect an EWS at the $q$-confidence level based on measuring $\psi$ at time $t$, we require that $\hat{\Delta}(t) > q(s(\hat{\psi}(t)) + s(\hat{\psi}_0))$, which, solved for $T_w$ gives for variance:

$$T_w > 2q^2 \left( \frac{\hat{\alpha}(t)/\sqrt{\hat{\alpha}_0} + \hat{\alpha}_0/\sqrt{\hat{\alpha}(t)}}{\hat{\alpha}_0 - \hat{\alpha}(t)} \right)^2, \qquad (7)$$

and for autocorrelation,

$$T_w > 2q^2 \left( \frac{\sqrt{\hat{\alpha}_0} + \sqrt{\hat{\alpha}(t)}}{\hat{\alpha}_0 - \hat{\alpha}(t)} \right)^2 \hat{\rho}_0^{-2}. \qquad (8)$$

Substituting $\alpha_0 = 2\sqrt{A|\lambda_0|}$ and $\alpha(t) = 2\sqrt{A|\lambda(t)|}$ provides the time window $T_w$ needed to detect an EWS at time $t$ with large probability. Eqs. (7) and (8) are illustrated in Fig. 4a (red and yellow curves), where it is seen that detecting a significant increase in variance requires a shorter data window than detecting a significant increase in autocorrelation. Two times $s(\hat{\psi}(t))$ around the mean of the ramped variance and two times $s(\hat{\psi}_0)$ around baseline values are illustrated in Fig. 4c, d (dashed lines). Once a trace leaves the baseline confidence interval, a statistically significant change is detected, and when the two dashed lines cross, 95% of the traces have detected an EWS (Eqs. (7) and (8)).

**Predicting a forthcoming collapse of the AMOC**
The AMOC fingerprint shown in Fig. 1e (replotted in Fig. 5a) shows an increased variance, $\gamma^2$, and autocorrelation, $\rho$, plotted in Fig. 5b, c as functions of the mid-point of a 50-year running window, i.e., the EWS obtained in 2020 is assigned to the year 1995. The estimates leave the confidence band of the baseline values (pink area) around the year 1970. This is not the estimate of $t_0$, which happened earlier and is still to be estimated; it is the year where EWSs are statistically different from baseline values. The estimates after 1970 stay consistently above the upper limit of the confidence interval and show an increasing trend, and we thus conclude that the system is moving toward the tipping point with high probability.

To estimate the tipping time once it has been established that the variance and autocorrelation are increasing, we use two independent methods to check the robustness of our results: (1) Moment-based estimator that uses the variance and autocorrelation estimates within the running windows. (2) Approximate MLE directly on model (1)-(2) with no running window. The advantage of the first method is that it has less model assumptions; however, it is sensitive to the choice of window size. The advantage of the second method is that it uses the information in the data more efficiently given model (1)-(2) is approximately correct, it has no need for a running window and does

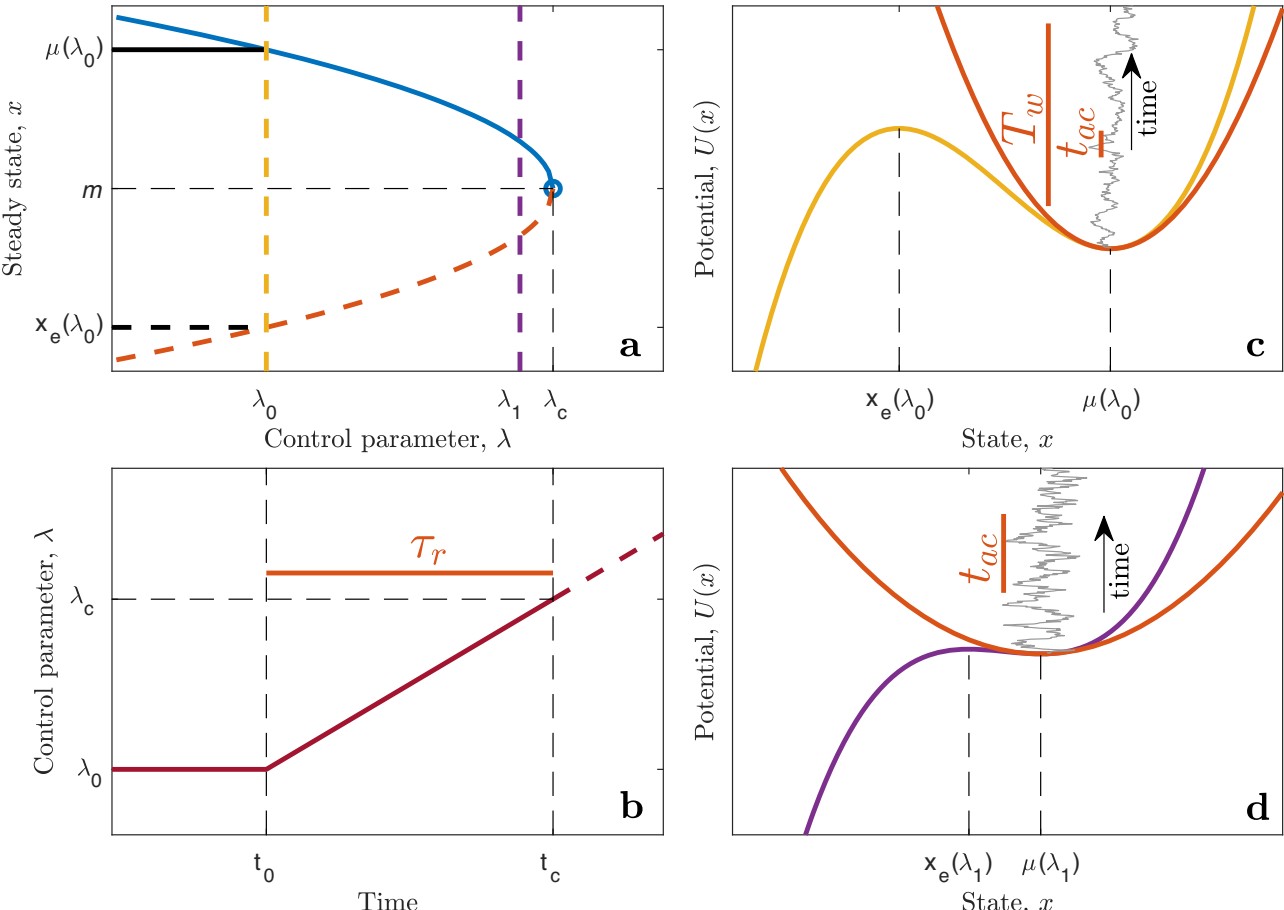

**Fig. 3 | Illustration of the transition and the time scales involved. a** Bifurcation diagram, the blue curve is the stable state (mean), $\mu$, and the dashed red curve is the unstable branch representing the edge state, $x_e$, both as functions of the control parameter $\lambda$. At $\lambda_c$ the system has a saddle-node bifurcation. **b** Linear ramping (Eq. (2)), where the system resides in a stationary state with $\lambda = \lambda_0$ until time $t_0$, from where $\lambda$ increases linearly with time and reach the critical value $\lambda_c$ at time $t_c$. **c** Effective dynamics (Eq. (1)) for $\lambda = \lambda_0$ indicated by the potential $U(x) = A(x-m)^3/3 + \lambda x$ (yellow) along the same color dashed line in (**a**). The red parabola is the

quadratic potential corresponding to the approximating linear Ornstein–Uhlenbeck (OU) process (Eq. (3)). The curve in gray is a realization of the OU process. The vertical short red bar shows the autocorrelation time, $t_{ac}$, for the process. The long vertical red bar symbolizes the time window, $T_w$ for calculating the early-warning signals (EWS). **d** As (**c**) for $\lambda_1$ close to $\lambda_c$ (purple dashed line in (**a**)). It can be seen directly that the gray realization has an increased autocorrelation and increased variance compared to the situation in (**c**).

not assume stationarity after time $t_0$. In general, MLE is statistically the preferred method of choice, giving the most accurate results with the lowest estimation variance.

The first method, the moment estimator of the tipping time obtains, within the running window, the parameters $\alpha(t)$ (Fig. 5d) and $\sigma^2$ (Fig. 5e) of the linearized dynamics, Eq. (3), and thus also $\gamma^2(t)$. Within the running window, the data are detrended before estimation by subtracting a linear regression fit in order not to falsely inflate the variance estimates caused by deviations from stationarity. Then we obtain $A\lambda(t)$ from $\sigma^2$ and $\gamma^2(t)$ (Fig. 5f) using that $A\lambda(t) = (\sigma^2/4\gamma^2(t))^2$. This is consistent with a linear ramping of $\lambda(t)$ beginning from a constant level $\lambda_0$ at a time $t_0$. By sweeping $t_0$ from 1910 to 1950 and $T_w$ from 45 to 65 years, we obtain $A\lambda_0$ and $\tau_r$ from the least square error fit to the data. This shows a single minimum at $t_0 = 1924$ and $T_w = 55$ years (Fig. 6e). Setting $t_0 = 1924$, we obtain $t_c$ from a linear fit (regressing $\lambda$ on $t$) from the crossing of the $x$-axis ($\lambda_c = 0$). This is shown in Fig. 5f (red line). This yields $-A\lambda_0 = 2.34$ year$^{-2}$ and $\tau_r = 133$ years. Thus, the tipping time is estimated to be in the year 2057, shown in Fig. 5f. Since we have only obtained the combined quantity $A\lambda = (\sigma^2/4\gamma^2)^2$, we still need to determine $A$ and $m$ in Eq. (1). We do that from the best linear fit to the mean level $\mu = m + \sqrt{|\lambda|/A}$ observing that $\mu = m + \sqrt{A|\lambda|}(1/A) = m + (\sigma^2/4\gamma^2)(1/A)$. The estimates are shown by the red curves in Fig. 5a–f. The red dot in Fig. 5a is the tipping point, and the dashed line in Fig. 5b

is the asymptote for the variance. With the parameter values completely determined, the confidence levels are calculated: The two-standard error levels around the baseline values of the EWS are shown by purple bands in Fig. 5b, c. Thus, both EWSs show increases beyond the two-standard error level from 1970 and onward.

The second method, the approximate MLE of the tipping time, is applied to model (1)–(2). The likelihood function is the product of transition densities between consecutive observations. However, the likelihood is not explicitly known for this model, and we therefore approximate the transition densities. From the data before time $t_0$, approximation (3) is used, where exact MLEs are available (Supplementary text S1). This provides estimates of the parameters $\lambda_0$, $m$ as a function of parameter $A$, as well as the variance parameter $\sigma^2$. To estimate $A$ and $\tau_r$, the observations after time $t_0$ are used. After time $t_0$, the linear approximation (3) is no longer valid because the dynamics are approaching the bifurcation point, and the non-linear dynamics will be increasingly dominating. The likelihood function is the product of transition densities, which we approximate with a numerical scheme, the Strang splitting, which has shown to have desirable statistical properties for highly non-linear models, where other schemes, such as the Euler–Maruyama approximation is too inaccurate[32] (Supplementary text S2). Using $t_0 = 1924$, the optimal fit is $t_c = 2065$, with a 95% confidence interval 2037–2109.

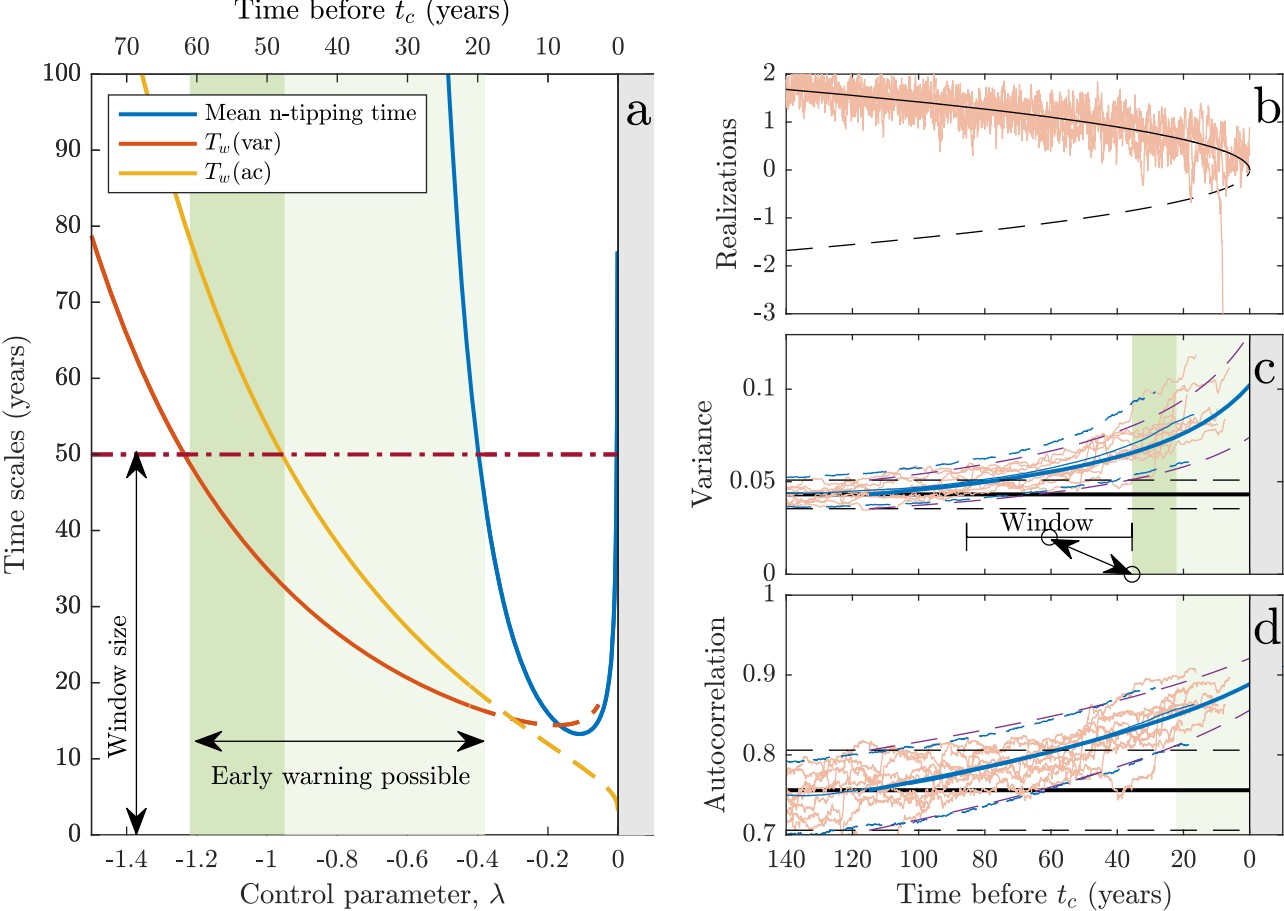

**Fig. 4 | Time scales. a** Time scales involved in the critical transition ramping the control parameter $\lambda$ from $\lambda_0 = -2.7$ to $\lambda_c = 0$, with a ramping time $\tau_r = 133$ years and $\sigma^2 = 0.3$. These parameters are obtained as best estimates from the Hadley Centre Sea Ice and Sea Surface Temperature data set (HadISST). The time remaining before $t_c$ is shown on top of the plot. The red and orange curves show the time window, $T_w$, needed in order to detect the increase in variance (red) and autocorrelation (orange) above the pre-ramping values at the 95% confidence level. Close to the bifurcation point, the (quasi-)stationarity approximation becomes less valid, which is indicated by the dashed part of the two curves. It is seen that detecting a significant increase in autocorrelation requires a longer data window than detecting a significant increase in variance. With $T_w = 50$ years (red dot-dashed line), an increase in variance can only be detected at the 95% confidence level after the red curve is below the 50-year level. The blue curve shows the mean waiting time for a noise-induced transition; when this becomes shorter than the 50-year level, the early-warning signal (EWS) is no longer relevant due to n-tipping occurring before $t_c$. Thus the range of time where an EWS can be applied is indicated by the green band (limited by the crossings of the red and blue curves with the size of the window). **b** Ten model realizations of the ramped approach to $t_c$, notice a few n-tippings prior to $t_c$. The black (black dashed) curve is the stable (unstable) fixed point of the model. **c** Increased variance as EWS: Black line is the pre-ramping steady state value, while the dashed lines are the two-standard error uncertainty range for calculating variance within the 50-year data window. The blue and dashed blue curves are the same but for the model approaching the transition. The brown curves correspond to the ten realizations in (**b**), while the green band corresponds to the green band in (**a**). The thin blue lines are the same obtained from simulating 1000 realizations. **d** Same as (**c**) but for the autocorrelation, where now the green band is narrower, corresponding to $T_{win}(ac)$ being smaller than the window size.

Confidence intervals for the estimate of the tipping time are obtained by bootstrap. The likelihood approach provides asymptotic confidence intervals; however, these assume that the likelihood is the true likelihood. To incorporate also the uncertainty due to the data generating mechanism (1) not being equal to the Ornstein–Uhlenbeck process (3) used in the likelihood, we chose to construct parametric bootstrap confidence intervals. This was obtained by simulating 1000 trajectories from the original model with the estimated parameters and repeating the estimation procedure on each data set. Empirical confidence intervals were then extracted from the 1000 parameter estimates. These were indeed larger than the asymptotic confidence intervals provided by the likelihood approach, however, not by much. Histograms of the bootstrapped estimates are shown in Fig. 6a–d. The histogram in Fig. 6a is the tipping year, repeated in yellow in 5f.

The mean of the bootstrapped estimates of the tipping time is $\langle t_c \rangle = 2065$, and the 95% confidence interval is 2037–2109. To test

the goodness-of-fit, normal residuals (see "Methods") were calculated for the data. These are plotted in Fig. 6f as a quantile-quantile plot. If the model is correct, the points fall close to a straight line. The model is seen to fit the data well, further supporting the obtained estimates.

## Discussion

We have provided a robust statistical analysis to quantify the uncertainty in observed EWSs for a forthcoming critical transition. The confidence depends on how rapidly the system is approaching the tipping point. With this, the significance of the observed EWSs for the AMOC has been established. This is a stronger result than just observing a significant trend in the EWS by, say, Kendall's $\tau$ test[27,33]. Here we calculate when the EWS are significantly above the natural variations. Furthermore, we have provided a method to not only determine whether a critical transition will happen but also an estimate of when it will happen. We predict with high confidence the tipping to happen as soon as mid-century (2037–2109 is a 95%

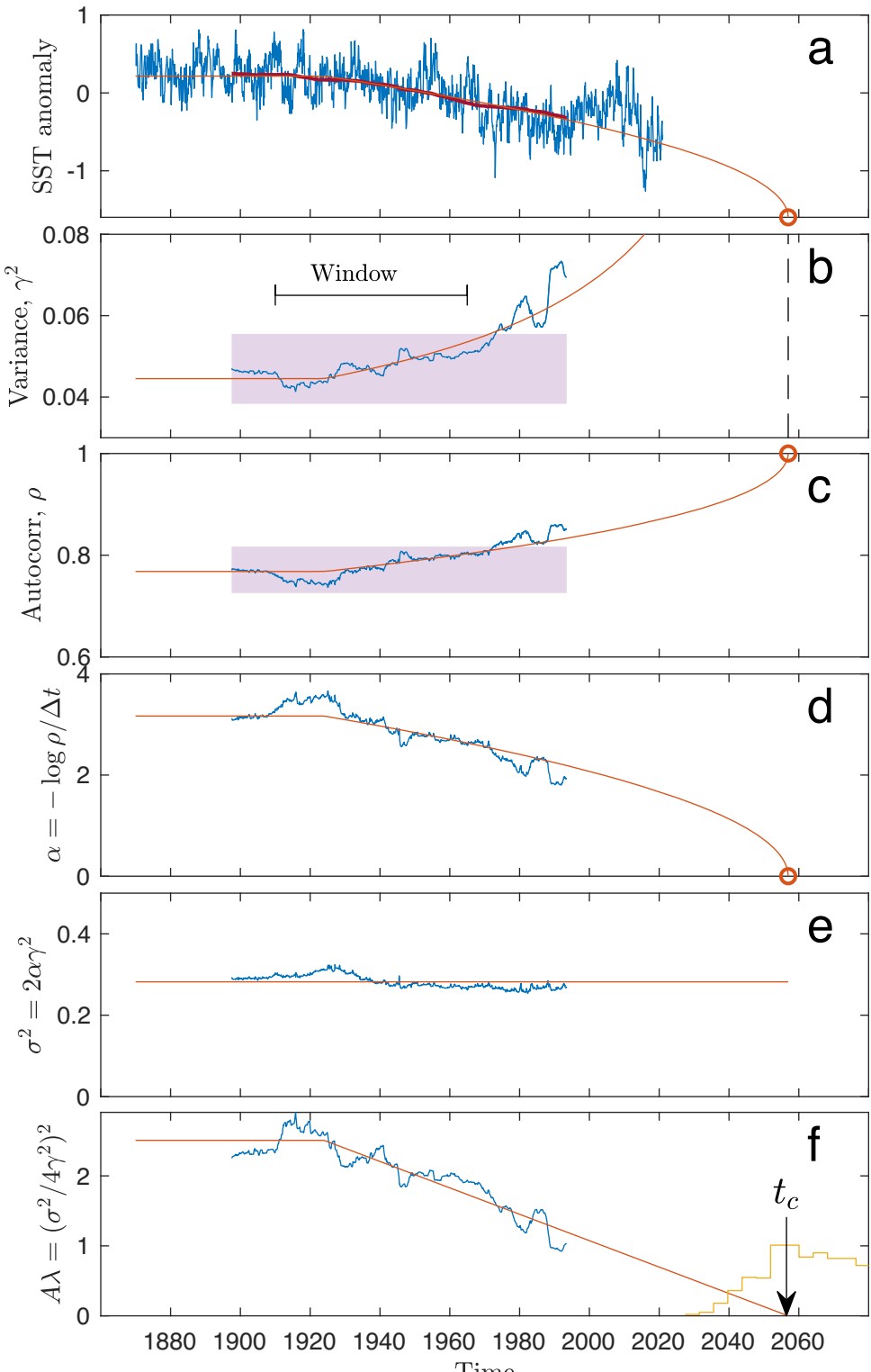

**Fig. 5 | Detection of early-warning signals and prediction of tipping time. a** Sea surface temperature (SST) anomaly (identical to Fig. 1e) together with the best estimate model of the steady state approaching a critical transition. **b**, **c** Variance and autocorrelation calculated within running 50-year windows, similar to Fig. 4c, d. The two-standard error level (indicated by the purple band) is obtained using the model to estimate the time varying $\alpha$ (**d**) and $\sigma^2$ (**e**) from the data. **f** Best estimate for $t_c$. The yellow histogram is the probability density for $t_c$ obtained by maximum likelihood estimates (see Supplementary Information S1 and S2).

confidence range). These results are under the assumption that the model is approximately correct, and we, of course, cannot rule out that other mechanisms are at play, and thus, the uncertainty is larger. However, we have reduced the analysis to have as few and sound assumptions as possible, and given the importance of the AMOC for

the climate system, we ought not to ignore such clear indicators of an imminent collapse.

The hysteresis simulations gathered in the model intercomparison[34] are equilibrium runs, for which a prediction of a future collapse does obviously not apply. Likewise, for the simulations

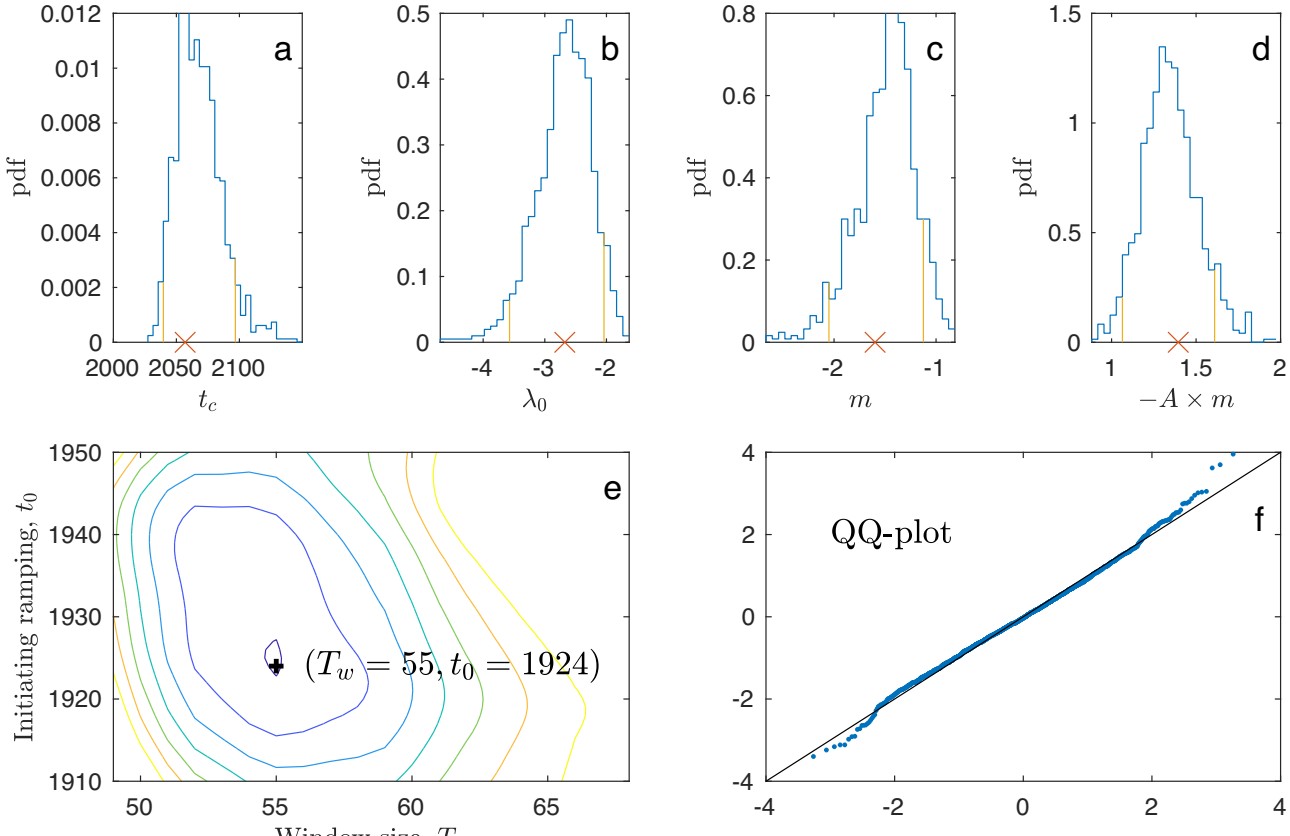

**Fig. 6 | Bootstrap confidence intervals, optimal estimation window and model control.** With parameters obtained from the data, a set of 1000 realizations of the model are used in a bootstrap study to assess the uncertainty of parameters. **a–d** Probability densities for $t_c$, $\lambda_0$, $m$ and $-A \times m$. Red crosses are the values obtained from the Atlantic meridional overturning circulation (AMOC) fingerprint data. The 95% confidence intervals are indicated by orange lines. The critical time $t_c$ is 2065, and the 95% confidence interval is 2037–2109. **e** Mean square error in fitting the ramping as a function of window size $T_w$ and time of initiating ramping, $t_0$. A unique minimum is found for $T_w = 55$ years and $t_0 = 1924$. **f** Quantile-quantile-plot of residuals from the model, if points fall close to a straight line (black line), the model fits the data well.

specified in the CMIP6 experiment. It could though be relevant to evaluate our method on state-of-the-art climate model simulations with linearly ramped external forcing and different ramping speeds in order to obtain the model-specific confidence in early prediction of the collapse judged solely from the EWSs.

Though we have established firm statistical methods to evaluate the confidence in the observed EWS, we can at present not rule out the possibility that a collapse will only be partial and not lead to a full collapse of the AMOC as suggested by some models: Note in Fig. 2, the "MPM" in the top panel and the "MOM hor" in the bottom panel both seem to show only partial tipping prior to the tipping to the complete shutdown of the AMOC. This result is also found in a more recent ocean model[35]. Furthermore, a high speed of ramping, i.e., a high speed at which the critical value of the control parameter is approached, could also increase the probability of tipping[36]. This scenario is the case of rate-induced tipping[37]. Even with these reservations, this is indeed a worrisome result, which should call for fast and effective measures to reduce global greenhouse gas emissions in order to avoid the steady change of the control parameter toward the collapse of the AMOC (i.e., reduce temperature increase and freshwater input through ice melting into the North Atlantic region). As a collapse of the AMOC has strong societal implications[38], it is important to monitor the flow and EWS from direct measurements[39–41].

## Methods

To obtain the AMOC fingerprint, two steps are required: The seasonal cycle in the SST is governed by the surface radiation independent from

the circulation and thus removed by considering the monthly anomalies, where the mean over the period of recording of the month is removed. Second, there is an ongoing positive linear trend in the SST related to global warming, which is also not related to circulation. This is compensated for by subtracting $2 \times$ the global mean (GM) SST anomaly (small seasonal cycle removed). This differs slightly from ref. 23, where $1 \times$ the GM SST was subtracted. The "translation" from the proxy SST temperature and AMOC flow is 0.26 SV/K [ref. 23, Fig. 3]. Here we have taken into account that the warming is not globally homogeneous: The warming in the SG region is larger than the global mean due to polar amplification. The way we have estimated this effect is by comparing the proxy with the AMOC estimates covering the period 1957–2004 from the so-called $MOC_z$ reported in the review by ref. 42. This shows a drop of 3 SV in that period. Minimizing the difference between the proxy $SST_{SG}$-$\beta$ $SST_{GM}$ and this more direct measurement with respect to $\beta$, we get $\beta = 1.95 \approx 2$ rather than $\beta = 1$ used by ref. 23. The factor 2 is thus the optimal value for the polar amplification[43] obtained by calibrating to recent direct measurements[42]. The original and our calibrated proxies are shown in Fig. 7.

To check the robustness with respect to the AMOC fingerprint record, we repeated the analysis, subtracting 1x and 3x GM SST from the SG SST. Subtracting 3x GM SST only changes estimate and the confidence intervals by a few years, whereas subtracting 1x GM SST delays the tipping with 58 years, but the overall trend and conclusions do not change. The results are given in Table 1. In the reanalysis, we fixed $t_0 = 1924$.

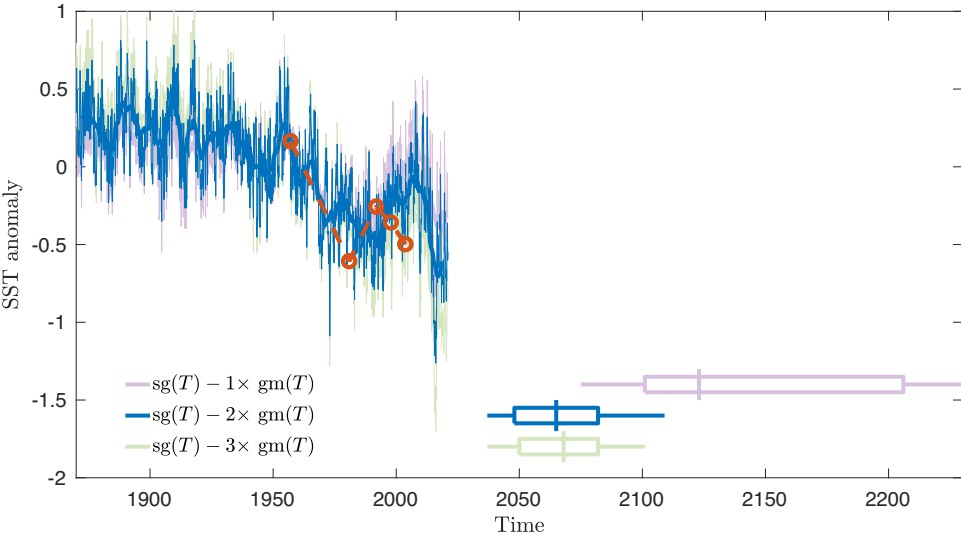

**Fig. 7 | Compensation of global warming in the Atlantic meridional overturning circulation (AMOC) fingerprint.** In the sea surface temperature (SST) AMOC fingerprint record, the compensation for global warming and polar amplification is done by subtracting the global SST ($SST_{GM}$) from the subpolar gyre (SG) SST ($SST_{SG}$). By calibrating by the $MOC_z$ AMOC proxy (red curves), the optimal AMOC proxy is $SST_{SG}$-2 $SST_{GM}$. To ensure the robustness of our results, we have repeated the analysis by subtracting 1x (purple) and 3x $SST_{GM}$ (green) and compared with the optimal 2x $SST_{GM}$ subtracted (blue). The corresponding estimates for the time of the collapse are shown in the same colors: The middle vertical line is the maximum likelihood estimate of the tipping time, the box represents the 66.6% confidence interval (Intergovernmental Panel on Climate Change (IPCC) definition of "likely"), while the horizontal line represents the 99% confidence intervals.

## Estimator of the tipping time and model control

The AMOC fingerprint is assumed to be observations from a process $X_t$ given as a solution to Eqs. (1) and (2), and we wish to estimate the parameters $\theta = (A, m, \lambda_0, \tau_r, \sigma)$ from observations $(x_0, x_1, ..., x_n)$ before time $t_0$ and observations $(y_0, y_1, ..., y_n)$ after time $t_0$. These equations cannot be explicitly solved, and the exact distribution of $X_t$ is not explicitly known. A standard way to solve this is to approximate the transition density by a Gaussian distribution obtained by the Euler–Maruyama scheme. However, the estimators obtained from the Euler–Maruyama pseudo-likelihood are known to be biased, especially in non-linear models[32]. We estimate by a two-step procedure, approximating the stationary distribution before time $t_0$ by an Ornstein–Uhlenbeck process, of which exact maximum likelihood estimators are available (see Supplementary text S1), and using Strang splitting for the non-stationary and non-linear part after time $t_0$, using methods proposed in ref. 32, see Supplementary text S2 for details.

To test the model fit, uniform residuals, $u_i, i = 1, ..., n$ were calculated for the AMOC data using the estimated parameters from the MLE method as follows. The model assumes that observation $x_i$ follows some distribution function $F_{i,\hat{\theta}}$ for the estimated parameter values $\hat{\theta}$. If this is true, then $u_i = F_{i,\hat{\theta}}(x_i)$ is uniformly distributed on (0, 1). Transforming these residuals back to a standard normal distribution provides standard normally distributed residuals if the model is true. Thus, a normal quantile-

quantile plot reveals the model fit. The points should fall close to a straight line. The reason for making the detour around the uniform residuals is twofold. First, since the data is not stationary, each observation follows its own distribution, and residuals cannot be directly combined. Second, since the model is stochastic, standard residuals are not well-defined, and observations should be evaluated according to their entire distribution, not only the distance to the mean.

## Noise-induced tipping

The drift term in Eq. (1) is the negative gradient of a potential, $f(x, \lambda) = -\partial_x V(x, \lambda) = -(A(x-m)^2 + \lambda)$ with $V(x, \lambda) = A(x-m)^3/3 + (x-m)\lambda$. For $\lambda < 0$, the drift has two fixed points, $m \pm \sqrt{|\lambda|/A}$. The point $m + \sqrt{|\lambda|/A}$ is a local minimum of the potential $V(x, \lambda)$ and is stable, whereas $m - \sqrt{|\lambda|/A}$ is a local maximum and unstable. The system thus has two basins of attraction separated by $m - \sqrt{|\lambda|/A}$, with a drift toward either $m + \sqrt{|\lambda|/A}$ or $-\infty$ dependent on whether $X_t > m - \sqrt{|\lambda|/A}$ or $X_t < m - \sqrt{|\lambda|/A}$. We denote the two basins of attraction, the normal and the tipped state, respectively. When $\lambda = 0$, the normal state disappears, and the system undergoes a bifurcation and $X_t$ will be drawn toward $-\infty$.

Due to the noise, the process can escape into the tipped state by crossing over the potential barrier $\Delta(\lambda) = V(m - \sqrt{|\lambda|/A}, \lambda) - V(m + \sqrt{|\lambda|/A}, \lambda) = 4|\lambda|^{3/2}/3A^{1/2}$. Assume $X_t$ to be close to $m + \sqrt{|\lambda|/A}$ at some time $t$, i.e., in the normal state. The escape time will asymptotically (for $\sigma \rightarrow 0$) follow an exponential distribution such that

$$P(t, \lambda) = 1 - \exp(-t/\tau_n(\lambda)) \quad (9)$$

where $P(t, \lambda)$ is the probability of observing an escape time shorter than $t$ for a given value of $\lambda$. The mean noise-induced escape time $\tau_n(\lambda)$ is[44,45]:

$$\tau_n(\lambda) = \frac{2\pi \exp(2\Delta(\lambda)/\sigma^2)}{\sqrt{V''(m + \sqrt{|\lambda|/A}, \lambda)|V''(m - \sqrt{|\lambda|/A}, \lambda)|}} \quad (10)$$
$$= (\pi/\sqrt{A|\lambda|}) \exp(8|\lambda|^{3/2}/3A^{1/2}\sigma^2).$$

**Table 1 | Estimates and confidence intervals for the tipping year using three proxies of the Atlantic meridional overturning circulation (AMOC), where the sea surface temperature (SST) is subtracted either 1, 2 or 3 times the global SST**

|  | Estimate | 95% CI | 66% CI |
|---|---|---|---|
| $SST_{SG}$-1 $SST_{GM}$ | 2123 | 2075–2288 | 2101–2206 |
| $SST_{SG}$-2 $SST_{GM}$ | 2065 | 2037–2109 | 2048–2082 |
| $SST_{SG}$-3 $SST_{GM}$ | 2068 | 2037–2101 | 2050–2082 |

Calibration with the $MOC_z$ AMOC proxy, the optimal is $SST_{SG}$-2 $SST_{GM}$, see Fig. 7.

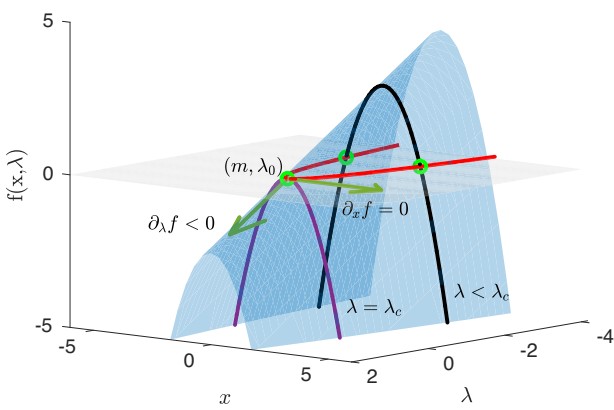

**Fig. 8 | Saddle-node bifurcation.** The function $f(x, \lambda)$ near a saddle point where a stable and an unstable fixed point merge at a saddle-node bifurcation. For $\lambda < \lambda_c$, there are two fixed points (green) where the black curve passes through the null plane, $f = 0$ (gray). The point in front is the stable fixed point, while the point in the back is the unstable fixed point. The red curve of fixed points is the bifurcation curve, with the stable branch in front and unstable branch in the back. The purple curve is $f(x, \lambda_c)$ which touches the null plane at one point $(x_0, \lambda_c)$. At this point, it is seen that $\partial_x f = 0$ and $\partial_\lambda f < 0$, indicated by the dark green tangents to the surface.

Assume that the rate of change of $\lambda(t)$ follows Eq. (2), then for $\tau_r < \tau_n(\lambda)$, the waiting time for a random crossing is so long that a crossing will not happen before a bifurcation-induced transition happens (b-tipping). If $\tau_r > \tau_n(\lambda)$, a noise-induced tipping is expected before the bifurcation point is reached. Since $\tau_n(\lambda)$ decreases with increasing $\lambda$, at some point, the two time scales will end up matching.

**Normal form of the saddle-node bifurcation**
Consider the general dynamical equation

$$\frac{dx}{dt} = f(x, \lambda), \tag{11}$$

where $x$ is a variable and $\lambda$ is a (fixed) parameter. A point $x_0$ with $f(x_0, \lambda) = 0$ is a fixed point or steady state. A fixed point is stable/unstable if $\partial_x f(x, \lambda)_{x = x_0}$ is negative/positive; thus, the fixed point is attracting/repelling. If $f(x, \lambda)$ is not a linear function of $x$, multiple steady states may exist. A saddle-node bifurcation occurs when changing the control parameter $\lambda$ through a critical value $\lambda_c$ a stable and an unstable fixed point merge and disappear. The situation is shown in the figure, where the blue surface is $f(x, \lambda)$, while the gray (null-) plane is $f(x, \lambda) = 0$. For a constant value of $\lambda$, the dynamics is determined by the black curve. The fixed points are determined by the intersection with the null plane (green); the point in the front is the stable fixed point, while the further point is the unstable fixed point. When changing $\lambda$ toward $\lambda_c = 0$, the two fixed points merge at the saddle-node bifurcation $(m, \lambda_c)$ (green). The normal form of the saddle node is obtained by expanding $f(x, \lambda)$ to the lowest order around the point $(m, \lambda_c)$, noting that $f(m, \lambda_c) = 0$, $\partial_x f(x, \lambda)_{(x, \lambda) = (m, \lambda_c)} = 0$ and $\partial_\lambda f(x, \lambda)_{(x, \lambda) = (m, \lambda_c)} < 0$ (see Fig. 8):

$$f(x, \lambda) \approx \frac{1}{2} \partial_{xx} f(x, \lambda)_{(x, \lambda) = (m, \lambda_c)} \times (x - m)^2 + \partial_\lambda f(x, \lambda)_{(x, \lambda) = (m, \lambda_c)}$$
$$\times (\lambda - \lambda_c) = -A(x - m)^2 - \tilde{\lambda}, \tag{12}$$

where $A = -\frac{1}{2} \partial_{xx} f(x, \lambda)_{(x, \lambda) = (m, \lambda_c)}$ and $\tilde{\lambda} = -\partial_\lambda f(x, \lambda)_{(x, \lambda) = (m, \lambda_c)} \times (\lambda - \lambda_c)$. This is the normal form for the saddle-node bifurcation. Thus, close to the bifurcation point, the stable steady state is

$$x_0 = m + \sqrt{-\tilde{\lambda}/A}. \tag{13}$$

In order to see that this is indeed the case for the AMOC transition also in comprehensive climate models, Fig. 2 is adapted from the model intercomparison study[34]. The steady state curves obtained are from simulations, with a very slowly changing control parameter (freshwater forcing). The top panel shows ocean-only models, while the bottom panel shows atmosphere-ocean models. The curves are, even away from the transition, surprisingly well fitted by Eq. (13). Note that for some models, the transition happens before the critical point, as should be expected from noise-induced transitions. Note also that the data has been smoothed such that increasing variance close to the transition is not visible. This observation strongly supports the assumption of a saddle-node bifurcation, while it also shows that $(m, \lambda_c)$ (black dots) are quite different between models, thus calling for reliable determination from observations.

## Data availability
Data can be found in the following repository: https://doi.org/10.17894/ucph.9ef73d8c-b642-4b55-88bc-b38984d043b9.

## Code availability
Computer code (Matlab and R) can be found in the following repository: https://doi.org/10.17894/ucph.9ef73d8c-b642-4b55-88bc-b38984d043b9.

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

## Acknowledgements

This work has received funding under the project Tipping Points in the Earth System (TiPES) from the European Union's Horizon 2020 research and innovation programme under grant agreement no. 820970. This is TiPES contribution #214. Furthermore, funding was provided by Novo Nordisk Foundation NNF20OC0062958; and European Union's Horizon 2020 research and innovation program under the Marie Skłodowska-Curie grant agreement No 956107, "Economic Policy in Complex Environments (EPOC)".

## Competing interests

The authors declare no competing interests.
