## [Peer Review File · Nature Communications]

Warning of a forthcoming collapse of the Atlantic meridional overturning circulationREVIEWER COMMENTS

Reviewer #1 (Remarks to the Author):

Review of: Warning of a forthcoming collapse of the Atlantic meridional overturning circulation, by Ditlevsen and Ditlevsen

I think this is an important paper, because it presents a methodological advance and because of the massive societal impacts if the analysis proves to be correct. I don't see anything fundamentally wrong with the methodology (with the caveat that I am no trained statistician). Nevertheless, there are statistical and systematic uncertainties in the approach. Hence, I would recommend publication after revisions, with more thorough discussion of uncertainties and more cautious wording of the findings.

For example, I think it is ill-advised to state "around 2057" for passing the tipping point, given the uncertainty and the past tendency of the popular media to hype AMOC-collapse findings. It would be better to cite a range in the abstract, i.e. 2034-2128 (95% range) or perhaps a "likely" range following the IPCC definition of likely (66.6%). Another way to summarize the finding would be to say that there is a significant risk of passing the tipping point already around mid-century, which is much earlier than many past studies and the IPCC have suggested. Stating the number 2057 suggests a precision that is not warranted and invites misleading headlines.

I recommend the authors discuss possible reasons for finding an earlier tipping point than in the CMIP5 or CMIP6 models reported in the IPCC reports, e.g. lack of meltwater forcing in these models, or systematic biases in Atlantic Ocean salinity in these models (as argued by Liu et al. 2017). Reasons why models may systematically overestimate AMOC stability have also been discussed by Hofmann & Rahmstorf 2009.

Specific comments:

Subtracting 2x the mean global warming obviously enhances the signal and is but a rough approximation of the SST warming in the subpolar Atlantic expected without AMOC decline. This is not unreasonable but presents a source of uncertainty. I would suggest the authors show the sensitivity of their results to this choice, perhaps simply by comparing 3 options, subtracting 1x, 2x and 3x the global mean SST to show how this affects the results.

A possible objection to this study is that the AMOC fingerprint time series is modelled with a simple dynamic with just one tipping point. Although it has been shown that 3D ocean GCMs overall tend to follow a similar dynamic (Fig. 2 and other studies), in some cases additional smaller bifurcations have been found in these models, where a transition in the AMOC occurs which does not lead to a complete shutdown (e.g. MPM and MOM hor curves in Fig. 2). Perhaps the analysis shows the proximity of such a more moderate transition, rather than a full shut-down? I don't know whether the authors have an argument that can rule this out; in any case this systematic uncertainty should be discussed.

Also, increased variance used as an EWS could also be affected by increasing variance in climatic drivers, rather than being due solely to approaching a tipping point. This caveat also needs to be discussed.

I am not sure how important the assumption of linear ramping up of λ is. If we consider global or NH temperature increase as driver of the control parameter change, it is not linear over the past 120 years. This may be another assumption leading to some systematic uncertainty.

A wording issue, page 4 near end: "we thus conclude that the system is approaching the tipping point with high probability". I think the authors possibly mean "is moving towards" rather than "is approaching" here? "Approaching" suggests the system is already close, as in "the train is approaching the station" which is only said when it is close but not during the whole journey during which it is moving towards the station all the time.

In the summary/discussion, it would be fitting to cite Michel et al. 2022, and also to discuss what is similar/different to Boers 2021, who arrived at a qualitatively similar conclusion.

References

Hofmann, M. and S. Rahmstorf (2009). "On the stability of the Atlantic meridional overturning circulation." *Proceedings of the National Academy of Sciences of the United States of America* 106(49): 20584-20589.

Liu, W., S.-P. Xie, Z. Liu and J. Zhu (2017). "Overlooked possibility of a collapsed Atlantic Meridional Overturning Circulation in warming climate." *Science Advances*: 7.

Michel, S. L. L., D. Swingedouw, P. Ortega, G. Gastineau, J. Mignot, G. McCarthy and M. Khodri (2022). "Early warning signal for a tipping point suggested by a millennial Atlantic Multidecadal Variability reconstruction." *Nat Commun* 13(1): 5176.

Reviewer #2 (Remarks to the Author):

Ditlevsen & Ditlevsen have created a very interesting and potentially very valuable piece of work by delving into the more of the statistics of EWS to provide more information than just seeing an increase in them to indicate a movement towards tipping. I have some suggestions that struck me whilst reading the paper and some more specific comments below.

My first comment is to wonder why the GCM outputs that are used to show that the normal saddle node bifurcation model is a great fit, are not run through the same analysis as the AMOC fingerprint to determine when they tip. This would be a great way to show the method in practice with some concrete results about the outcome. If I have missed a reason why this cannot be done then it would be good to be more explicit about this in the manuscript.

My other main comments concern the practicality of this method. Could you for example, link this to GMT instead of time (as is mentioned as a potential main driver in the introduction), in which case you could determine a threshold temperature value not to cross (providing it is not too wide and useful). I would also like some discussion on the potential for using this for other systems (e.g. how likely is it that they follow the same norm. form bifurcation model? Or how easy is it to transfer the method to other systems and how much do you need to know about them?).

At the moment the manuscripts feel a bit too statistical, and the suggestions above should make it more balanced towards results and applications if they are valid points.

I have a few specific points below, alongside some more general language comments which I will withhold until a new version of the manuscript.

In equation (1), μ does not appear yet the text that follows suggests that it does.

This might be me misunderstanding, but are the 2-sigma bands in Fig. 4 panels actually related to the increasing indicators rather than the baseline? Reading the manuscript, I got the impression that the baseline uncertainties could be made arbitrarily small by varying window size, and that the uncertainty bounds on the indicators were used to measure statistical significance. Perhaps this just needs a better explanation, as I say it is probably my misunderstanding.

Figure 5 is not really mentioned in the paper, I assume it should be linked to the MLE calculations. Only 5b is mentioned at all.

SI figures are labelled Figs 6 & 7 rather than S1 & S2. I'm not aware of the format for the journal but worth checking if this is the correct way to do it.

Reviewer #3 (Remarks to the Author):

What are the noteworthy results?

The paper uses EWS to predict an upcoming tipping point in the AMOC. The authors use two methods to detect a transition: 1) moment based variance and autocorrelation estimates, and 2) a maximum likelihood estimator. They find that a collapse occurs around 2057 with 95% confidence interval 2034-2128.

Will the work be of significance to the field and related fields? How does it compare to the established literature? If the work is not original, please provide relevant references.

The work is of significance to both the Theoretical EWS community as well as climate change scientists. The AMOC has long been identified as a potential tipping element in the climate system, but I am not aware of any other studies that provide an estimate of when a tipping point might occur.

The study cites most relevant literature when it comes to EWS and tipping points in simplistic models. However, the study is not linking to tipping points in the AMOC in larger scale climate model. I think the study could be strengthened by either making this link very explicit, or even repeating their analysis on model output.

Does the work support the conclusions and claims, or is additional evidence needed?

In the current state, I think the manuscript is overstating the reliability and implications of its results. Specifically: there is a complete lack of any critical reflection on the implications of the assumptions of the methodology. For instance, the authors assume a linear increase in some unknown control parameter. The estimation of the time of tipping, as well as the confidence intervals, are dependent on this assumption of linearity. I agree with the authors that without any knowledge, the linear assumptions makes sense. However, this does affect the interpretation of the confidence intervals, and I believe the authors have the responsibility to be more transparent about this.

Are there any flaws in the data analysis, interpretation and conclusions? - Do these prohibit publication or require revision?

As far as I can see there are no flaws in the data analysis, but the interpretation and conclusions are overstated and should be reformulated.

Is the methodology sound? Does the work meet the expected standards in your field? Is there enough detail provided in the methods for the work to be reproduced?

The methodology is sound and the analysis is in principle reproducible.

Some other minor issues:

- The abstract is currently not representative of the manuscript, the abstract focuses too much on the statistical analysis and robustness of EWS, and not on the timing of the collapse of the AMOC.
- In the abstract, the authors mention the 'business as usual scenario'. Does this refer to RCP 8.5? This seems oddly specific and does not come back in the article.
- The steps in the methodology are quite difficult to understand for non-mathematicians. To reach a wide audience, it would greatly help to add some visualizations of the steps of the methodology (to the supplementary materials?).

REVIEWER COMMENTS

Reviewer #1 (Remarks to the Author):

Review of: Warning of a forthcoming collapse of the Atlantic meridional overturning circulation, by Ditlevsen and Ditlevsen

I think this is an important paper, because it presents a methodological advance and because of the massive societal impacts if the analysis proves to be correct. I don't see anything fundamentally wrong with the methodology (with the caveat that I am no trained statistician). Nevertheless, there are statistical and systematic uncertainties in the approach. Hence, I would recommend publication after revisions, with more thorough discussion of uncertainties and more cautious wording of the findings.

For example, I think it is ill-advised to state "around 2057" for passing the tipping point, given the uncertainty and the past tendency of the popular media to hype AMOC-collapse findings. It would be better to cite a range in the abstract, i.e. 2034-2128 (95% range) or perhaps a "likely" range following the IPCC definition of likely (66.6%). Another way to summarize the finding would be to say that there is a significant risk of passing the tipping point already around mid-century, which is much earlier than many past studies and the IPCC have suggested. Stating the number 2057 suggests a precision that is not warranted and invites misleading headlines.

This is a very good point, thanks. We have decided to write the most cautious "around mid-century" in the Abstract and the more explicit "2021-2079 (95% confidence range)" in the main text.

I recommend the authors discuss possible reasons for finding an earlier tipping point than in the CMIP5 or CMIP6 models reported in the IPCC reports, e.g. lack of meltwater forcing in these models, or systematic biases in Atlantic Ocean salinity in these models (as argued by Liu et al. 2017). Reasons why models may systematically overestimate AMOC stability have also been discussed by Hofmann & Rahmstorf 2009.

Thanks for pointing this out. These are relevant references, which we have added together with two more in a new paragraph discussing the ipcc assessment and the bias in the stability of the AMOC in CMIP5 and CMIP6 models.

Specific comments:

Subtracting 2x the mean global warming obviously enhances the signal and is but a rough approximation of the SST warming in the subpolar Atlantic expected without AMOC decline. This is not unreasonable but presents a source of uncertainty. I would suggest the authors show the sensitivity of their results to this choice, perhaps simply by comparing 3 options, subtracting 1x, 2x and 3x the global mean SST to show how this affects the results.

This is a good point, and in fact, we had already done these calculations. The results are relatively robust, in the sense that the collapse can with high probability be expected to happen within the next half century. A sentence has been added in the main text, and the results are now reported in Section S6, where the following table is included:

	Estimate	95% CI	66% CI
SST - 1 global SST	2082	2046 - 2170	2064 - 2124
SST - 2 global SST	2057	2021 - 2079	2038 - 2062
SST - 3 global SST	2057	2026 - 2077	2038 - 2063

A possible objection to this study is that the AMOC fingerprint time series is modelled with a simple dynamic with just one tipping point. Although it has been shown that 3D ocean GCMs overall tend to follow a similar dynamic (Fig. 2 and other studies), in some cases additional smaller bifurcations have been found in these models, where a transition in the AMOC occurs which does not lead to a complete shutdown (e.g. MPM and MOM hor curves in Fig. 2). Perhaps the analysis shows the proximity of such a more moderate transition, rather than a full shut-down? I don't know whether the authors have an argument that can rule this out; in any case this systematic uncertainty should be discussed.

This is a valid point. We have added a discussion to the last section, which we renamed (as required by the journal) "Discussion": It is fair to say that from an analysis of observations in the present state of the AMOC, we cannot rule out that passing the tipping point only leads to a partial shutdown. In order to do that, we have to rely on models, which at the present stage give too diverging results to base firm predictions on. We include this caveat and beside noting the diverging behavior of the "MPM" and "MOM hor" models (thanks for explicitly pointing this out). Furthermore, we refer to very recent model studies by Lohmann et al. (including one of us!) with similar behavior. Another important issue, which we also add to the discussion is the dependence of the rate of change in the risk of tipping. This point also relates to the comment below and the comment by reviewer #3 on the assumption of a linear ramping.

Also, increased variance used as an EWS could also be affected by increasing variance in climatic drivers, rather than being due solely to approaching a tipping point. This caveat also needs to be discussed.

Here we are not quite sure if we agree with the reviewer. It is of course a possibility that the observed EWS by some strange coincidences would be a response to the changes in the climatic drivers. There are two arguments against such a scenario: Firstly, we do believe that the main driver of climate change is the increasing greenhouse gas concentration in the atmosphere. This driver does (sadly) show an increase, but it does not show an increased variance (enough to explain the EWS). Secondly, this remark touches upon a discussion on the consistency of EWS: In order to attribute increase in autocorrelation as an EWS, it must be accompanied with an increase in variance (and vice versa), if this is not consistent, one should worry about such a wrong attribution of increased variance as an EWS. Such a check is inherent in our analysis. We thus do not believe there is a caveat. We could add such a discussion, but we judge that this would be an unnecessary technical complication added to the paper.

I am not sure how important the assumption of linear ramping up of lambda is. If we consider global or NH temperature increase as driver of the control parameter change, it is not linear over the past 120 years. This may be another assumption leading to some systematic uncertainty.

Assuming a linear ramping is the best “uninformed”, with no explicit consideration on the control parameter. We agree that global (or NH) temperature increase is not linear. We do not assume this to be the “driver”. However, if one should insist on a “driver”, we find that the greenhouse forcing $\sim \log([\text{CO}_2])$ would be the natural choice. This has indeed increased rather linearly since the beginning of monitoring in 1957 (Keeling curve), and can safely be assumed to have done so since the beginning of the industrial period. We have made this point explicitly with reference to

C. D. Keeling, S. C. Piper, R. B. Bacastow, M. Wahlen, T. P. Whorf, M. Heimann, and H. A. Meijer, Exchanges of atmospheric CO₂ and ¹³CO₂ with the terrestrial biosphere and oceans from 1978 to 2000. I. Global aspects, SIO Reference Series, No. 01-06, Scripps Institution of Oceanography, San Diego, 88 pages, 2001. <http://escholarship.org/uc/item/09v319r9>

A wording issue, page 4 near end: “we thus conclude that the system is approaching the tipping point with high probability”. I think the authors possibly mean “is moving towards” rather than “is approaching” here? “Approaching” suggests the system is already close, as in “the train is approaching the station” which is only said when it is close but not during the whole journey during which it is moving towards the station all the time.

We have revised to “is moving towards”. English is not our native language, we cannot judge the subtle difference, so thank you for pointing this out.

In the summary/discussion, it would be fitting to cite Michel et al. 2022, and also to discuss what is similar/different to Boers 2021, who arrived at a qualitatively similar conclusion.

Yes, done!

References

Hofmann, M. and S. Rahmstorf (2009). "On the stability of the Atlantic meridional overturning circulation." *Proceedings of the National Academy of Sciences of the United States of America* 106(49): 20584-20589.

Liu, W., S.-P. Xie, Z. Liu and J. Zhu (2017). "Overlooked possibility of a collapsed Atlantic Meridional Overturning Circulation in warming climate." *Science Advances*: 7.

Michel, S. L. L., D. Swingedouw, P. Ortega, G. Gastineau, J. Mignot, G. McCarthy and M. Khodri (2022). "Early warning signal for a tipping point suggested by a millennial Atlantic Multidecadal Variability reconstruction." *Nat Commun* 13(1): 5176.

Reviewer #2 (Remarks to the Author):

Ditlevsen & Ditlevsen have created a very interesting and potentially very valuable piece of work by delving into the more of the statistics of EWS to provide more information than just seeing an increase in them to indicate a movement towards tipping. I have some suggestions that struck me whilst reading the paper and some more specific comments below.

My first comment is to wonder why the GCM outputs that are used to show that the normal saddle node bifurcation model is a great fit, are not run through the same analysis as the AMOC fingerprint to determine when they tip. This would be a great way to show the method in practice with some concrete results about the outcome. If I have missed a reason why this cannot be done then it would be good to be more explicit about this in the manuscript.

This is a very valid point, and we did look through model repositories. Most model simulations establishing the hysteresis are not ramping experiments, they are run to equilibrium before incrementally changing the freshwater forcing, thus a temporal prediction is not relevant. If there are ramping experiments, it would require variables representing the AMOC to be stored at sufficient resolution. The most relevant way to show the method on GCM simulations, would be to run specifically designed to pass the saddle-node bifurcation with a few different ramping rates. This is worth considering as a future project. We have added a discussion of this to the Summary and Discussion section.

My other main comments concern the practicality of this method. Could you for example, link this to GMT instead of time (as is mentioned as a potential main driver in the introduction), in

which case you could determine a threshold temperature value not to cross (providing it is not too wide and useful). I would also like some discussion on the potential for using this for other systems (e.g. how likely is it that they follow the same norm. form bifurcation model? Or how easy is it to transfer the method to other systems and how much do you need to know about them?).

We could indeed plot the EWSs against GMT. However, as discussed above, we find that it is more robust not to consider GMT or NH temperature as the control parameter, but rather only assume a linear ramping. With respect to the prediction of the bifurcation point we can then only assume a linear extrapolation.

The method is general for any system showing a saddle-node bifurcation, so yes, the method is indeed useful for other systems. We are applying the method to other time series from both ecological and medical systems believed to be moving towards a saddle node bifurcation, and indeed, the method works more generally (this is for a different article). Nothing is needed to be known about these systems, since the method first determines whether EWS's are present with statistical confidence, then the tipping time is estimated, and finally, model control is performed on uniform residuals to evaluate if the model can be assumed to explain the data or not.

At the moment the manuscripts feel a bit too statistical, and the suggestions above should make it more balanced towards results and applications if they are valid points.

Yes, we believe that our added discussions helps that balance.

I have a few specific points below, alongside some more general language comments which I will withhold until a new version of the manuscript.

Thanks for your willingness to go over our language.

In equation (1), μ does not appear yet the text that follows suggests that it does.

We have reformulated.

This might be me misunderstanding, but are the 2-sigma bands in Fig. 4 panels actually related to the increasing indicators rather than the baseline? Reading the manuscript, I got the impression that the baseline uncertainties could be made arbitrarily small by varying window size, and that the uncertainty bounds on the indicators were used to measure statistical significance. Perhaps this just needs a better explanation, as I say it is probably my misunderstanding.

This is our mistake! Indeed, the text said that the baseline uncertainties could be made arbitrarily small, but we corrected that realizing that also the baseline is determined with uncertainties, since it is unrealistic to assume infinite time series (and indeed, we do not have that for the AMOC). Thus, in old Fig. 4, now Fig. 5, the 2-sigma uncertainty levels are for the

baseline values. We apologize for the confusion, all calculations and formulas are correct, and eqs. (7) and (8) are using the uncertainty also in the baseline estimates. We had some old text hanging from the working process. We corrected the text, and made some clarifications.

Figure 5 is not really mentioned in the paper, I assume it should be linked to the MLE calculations. Only 5b is mentioned at all.

Figure 5 is now Figure 6. Figure 6, panels a-d and f are now referenced in section "Uncertainty in the estimate of the tipping time" and panel e is referenced in section "1. Moment estimator of the tipping time".

SI figures are labelled Figs 6 & 7 rather than S1 & S2. I'm not aware of the format for the journal but worth checking if this is the correct way to do it.

Changed to figures S1 and S2.

Reviewer #3 (Remarks to the Author):

What are the noteworthy results?

The paper uses EWS to predict an upcoming tipping point in the AMOC. The authors use two methods to detect a transition: 1) moment based variance and autocorrelation estimates, and 2) a maximum likelihood estimator. They find that a collapse occurs around 2057 with 95% confidence interval 2034-2128.

Will the work be of significance to the field and related fields? How does it compare to the established literature? If the work is not original, please provide relevant references.

The work is of significance to both the Theoretical EWS community as well as climate change scientists. The AMOC has long been identified as a potential tipping element in the climate system, but I am not aware of any other studies that provide an estimate of when a tipping point might occur.

The study cites most relevant literature when it comes to EWS and tipping points in simplistic models. However, the study is not linking to tipping points in the AMOC in larger scale climate model. I think the study could be strengthened by either making this link very explicit, or even repeating their analysis on model output.

This is a good point, also raised by reviewer #2, thus we refer to our response above.

Does the work support the conclusions and claims, or is additional evidence needed?

In the current state, I think the manuscript is overstating the reliability and implications of its results. Specifically: there is a complete lack of any critical reflection on the implications of the

assumptions of the methodology. For instance, the authors assume a linear increase in some unknown control parameter. The estimation of the time of tipping, as well as the confidence intervals, are dependent on this assumption of linearity. I agree with the authors that without any knowledge, the linear assumptions makes sense. However, this does affect the interpretation of the confidence intervals, and I believe the authors have the responsibility to be more transparent about this.

Here the reviewer identifies Achilles' heel in any calculation of confidence intervals. When assuming a specific model (eq. 1) it is possible to calculate confidence, however, it is not possible to calculate the reliability of the model. This has to be judged based on other types of reasoning. In our view, the important point is to avoid unjustified assumptions and in general make as few and sound assumptions as possible. We believe that the discussions of the caveats and the assumption of a linear ramping has been included as a response to reviewers #1 and #2. We also added further discussion on the confidence of the results in the "Summary and Discussion" section.

Are there any flaws in the data analysis, interpretation and conclusions? - Do these prohibit publication or require revision?

As far as I can see there are no flaws in the data analysis, but the interpretation an conclusions are overstated and should be reformulated.

Done, see also suggestions by reviewer #1.

Is the methodology sound? Does the work meet the expected standards in your field? Is there enough detail provided in the methods for the work to be reproduced?

The methodology is sound and the analysis is in principle reproducible.

Some other minor issues:

- The abstract is currently not representative of the manuscript, the abstract focuses too much on the statistical analysis and robustness of EWS, and not on the timing of the collapse of the AMOC.

We have changed the Abstract to be more representative. We also shortened it considerably to comply with Journal requirements.

- In the abstract, the authors mention the 'business as usual scenario'. Does this refer to RCP 8.5? This seems oddly specific and does not come back in the article.

We did not have the RCPs, or present SSPs in mind when referring to 'business as usual'. This was deliberately vague, since what we really assume is the linear-in-time approach to the tipping point. Since we do not assume the driver known, it would in our view not be warranted to refer to a specific future RCP scenario. We have slightly rephrased the sentence in the abstract.

- The steps in the methodology are quite difficult to understand for non-mathematicians. To reach a wide audience, it would greatly help to add some visualizations of the steps of the methodology (to the supplementary materials?).

We have based on the suggestion made an additional figure (Figure 3) illustrating the scenario and explaining the time scales involved. We have added it to the main manuscript and hope that this does not conflict with requirements on length.

Again, we will take the opportunity to thank the reviewers for very constructive criticism and suggestions, which we believe have resulted in a much improved revision.

Best regards

Peter Ditlevsen and Susanne Ditlevsen.

REVIEWERS' COMMENTS

Reviewer #1 (Remarks to the Author):

I am satisfied by the authors' revisions and recommend publication.

Minor wording issues:

Is it called "critical slow down" or "critical slowing down"?

And in the Discussion: "a high speed of ramping, i.e. a high speed at which the critical value of the control parameter is reached, could also influence the probability of tipping": "influence" is neutral in which direction. Isn't it the case that it can increase the probability? Then say that. And should it say "approached" rather than "reached" here?

Reviewer #2 (Remarks to the Author):

I find the updated version of the manuscript to be much better, particularly the new Fig. 3 which is very helpful in understanding the parameters and concepts visually. I thank the authors for taking my original comments on board and am happy that they have been addressed. I have found some very minor typos which I have listed below but am otherwise happy with this interesting piece of work:

Page 1: Should be 'besides' after reference 17. Also EWS on the line below has not been mentioned yet (I assume this should be in the abstract).

Page 2: There is a capital 'A' on 'area' when I don't think there should be. I also think near the end of the first paragraph the sentence should read '...and only changes the confidence intervals by a few years...'

Page 5: In the paragraph before eq 3 reads better as '...thus, a shorter window T_w is required for detecting...' I think.

Figure 4: Caption says 'HasISST' rather than 'HadISST'.

Page 8: There's a space missing before 'The standard error'. Also 'The first/second method' after '1./2.' seems a bit redundant but is personal choice.

Figure 5: Panel labels are missing.

Page 12: 'asmid-century'.

Reviewer #3 (Remarks to the Author):

The authors have addressed all of my concerns.

As I am neither a statistician nor a mathematician, I am still having trouble wrapping my head about the methodological steps for the maximum likelihood estimator, despite the figure added by the authors. Conceptually however, it all makes sense.

Furthermore, with the new tone in the writing and more cautious interpretation of the results, I can support the publication of this work.

Response to reviewers, second revision, Nature Communications

Thank you for your positive feedback. Here we answer point by point, first to the reviewers, then to the author checklist.

All changes in the manuscript are marked in red.

REVIEWERS' COMMENTS

Reviewer #1 (Remarks to the Author):

I am satisfied by the authors' revisions and recommend publication.

Minor wording issues:

Is it called "critical slow down" or "critical slowing down"?

Answer: It is called "critical slowing down". Thank you for noticing, we have changed throughout.

And in the Discussion: "a high speed of ramping, i.e. a high speed at which the critical value of the control parameter is reached, could also influence the probability of tipping":

"influence" is neutral in which direction. Isn't it the case that it can increase the probability?

Then say that. And should it say "approached" rather than "reached" here?

*Answer: We changed accordingly to: "a high speed at which the critical value of the control parameter is **approached**, could also **increase** the probability of tipping".*

Reviewer #2 (Remarks to the Author):

I find the updated version of the manuscript to be much better, particularly the new Fig. 3 which is very helpful in understanding the parameters and concepts visually. I thank the authors for taking my original comments on board and am happy that they have been addressed. I have found some very minor typos which I have listed below but am otherwise happy with this interesting piece of work:

Page 1: Should be 'besides' after reference 17. Also EWS on the line below has not been mentioned yet (I assume this should be in the abstract).

Answer: Corrected.

Page 2: There is a capital 'A' on 'area' when I don't think there should be. I also think near the end of the first paragraph the sentence should read '...and only changes the confidence intervals by a few years...'

Answer: Corrected.

Page 5: In the paragraph before eq 3 reads better as '...thus, a shorter window T_w is required for detecting...' I think.

Answer: Corrected.

Figure 4: Caption says 'HasISST' rather than 'HadISST'.

Answer: Corrected.

Page 8: There's a space missing before 'The standard error'. Also 'The first/second method' after '1./2.' seems a bit redundant but is personal choice.

Answer: Corrected. We agree that the repetitions are redundant, we removed it.

Figure 5: Panel labels are missing.

Answer: Corrected; new figure uploaded.

Page 12: 'asmid-century'.

Answer: Corrected.

Reviewer #3 (Remarks to the Author):

The authors have addressed all of my concerns.

As I am neither a statistician nor a mathematician, I am still having trouble wrapping my head about the methodological steps for the maximum likelihood estimator, despite the figure added by the authors. Conceptually however, it all makes sense.

Furthermore, with the new tone in the writing and more cautious interpretation of the results, I can support the publication of this work.